# Mechanical Characteristics Evaluation of a Single Ply and Multi-Ply Carbon Fiber-Reinforced Plastic Subjected to Tensile and Bending Loads

**DOI:** 10.3390/polym14153213

**Published:** 2022-08-07

**Authors:** Hadăr Anton, Baciu Florin, Voicu Andrei-Daniel, Vlăsceanu Daniel, Tudose Daniela-Ioana, Adetu Cătălin

**Affiliations:** 1Department of Strength of Materials, Faculty of Industrial Engineering and Robotics, University Politehnica of Bucharest, 313 Splaiul Independenței, Sector 6, 060042 Bucharest, Romania; 2Academy of Romanian Scientists, 3 Ilfov Street, Sector 5, 050045 Bucharest, Romania; 3Technical Sciences Academy of Romania, 26 Dacia Boulevard, Sector 1, 030167 Bucharest, Romania

**Keywords:** carbon fiber reinforced composite, digital image correlation, tensile test, flexural test, mechanical behavior, mechanical characteristics determination

## Abstract

Carbon fiber-reinforced composites represent a broadly utilized class of materials in aeronautical applications, due to their high-performance capability. The studied CFRP is manufactured from a 3K carbon biaxial fabric 0°/90° with high tensile resistance, reinforced with high-performance thermoset molding epoxy vinyl ester resin. The macroscale experimental characterization has constituted the subject of various studies, with the scope of assessing overall structural performance. This study, on the other hand, aims at evaluating the mesoscopic mechanical behavior of a single-ply CFRP, by utilizing tensile test specimens with an average experimental study area of only 3 cm^2^. The single-ply tensile testing was accomplished using a small scale custom-made uniaxial testing device, powered by a stepper motor, with measurements recorded by two 5-megapixel cameras of the DIC Q400 system, mounted on a Leica M125 digital stereo microscope. The single-ply testing results illustrated the orthotropic nature of the CFRP and turned out to be in close correlation with the multi-ply CFRP tensile and bending tests, resulting in a comprehensive material characterization. The results obtained for the multi-ply tensile and flexural characteristics are adequate in terms of CFRP expectations, having a satisfactory precision. The results have been evaluated using a broad experimental approach, consisting of the Dantec Q400 standard digital image correlation system, facilitating the determination of Poisson’s ratio, correlated with the measurements obtained from the INSTRON 8801 servo hydraulic testing system’s load cell, for a segment of the tensile and flexural characteristics determination. Finite element analyses were realized to reproduce the tensile and flexural test conditions, based on the experimentally determined stress–strain evolution of the material. The FEA results match very well with the experimental results, and thus will constitute the basis for further FEA analyses of aeronautic structures.

## 1. Introduction

Composite materials represent the technological leading edge in terms of structural optimization for all major industrial domains, as a consequence of rigorous demands for better strength and mass properties. Although the manufacturing costs are slightly higher than those of traditional metal alloys [1], the benefits are without a doubt worth the financial effort thanks to their major advantages. Automotive engineering [2], marine structures [3], infrastructure works [4] and more recently the additive manufacturing industry [5] are all playing a part in developing improved composite materials, to meet the demands of increasingly challenging tasks. The aircraft manufacturing industry is undoubtedly one of the major investors in composite development. Over the past years, composite components found their way into nearly every corner of an aircraft, starting with small components, such as bulkheads or stringers [6], and ending with massive fuselage parts [7]. One important aspect is the manufacturing processes diversification, which occurred over the past decades [8], opening the market for middle range manufacturers, thus making composite materials an affordable solution for a broader range of customers. Even more delicate structures that are considered critical for aircraft safety, such as wings [9] or compressor blades [10], are now being considered for a composite conversion, due to the increasing confidence in their capabilities. A few of the many advantages that composites possess over traditional materials are the ability to tailor properties accordingly with the performance demands, an excellent strength-to-weight ratio and not the least, the chemical characteristics (resistance to corrosion and chemical resistance) [11]. Metal aircraft components are especially considered for a composite material upgrade due to the light weight they possess and also due to certain important capabilities, such as on-site manufacturing/repairing or the lower degree of maintenance they imply [12].

Composites are manufactured by combining two or more materials with exceptional properties, which also possess a high degree of compatibility. In order to prove their worthiness to fulfill difficult tasks, composites have to undergo a series of experimental tests, depending on the load type and direction. Tensile or compression testing, shear testing, bending or fatigue testing are only a few examples of the investigations that must be covered in order to obtain a complete set of mechanical characteristics [13]. The macro-scale tensile testing effects on carbon fiber-reinforced plastics have been intensely studied during the last decade [14,15], with the inconvenience of allowing only the assessment of the overall material deformation. In order to have a more precise understanding of the large-scale effects, it is imperative to observe deformations that occur on the smallest scale possible, in this case being a single ply of a laminated composite. The microscale fracture mechanics also constitutes a frequently approached subject [16], shedding light on the failure emergence and propagation.

Digital image correlation is an optical technique, which was developed in the middle 1970s, that has recently recaptured the interest of scientists involved in mechanical applications research [17,18]. One of the major advantages of this procedure is that it does not imply any direct contact with the object being studied, thus eliminating the necessity of additional equipment, such as strain gages or extensometers, and yet providing a full range of mechanical characteristics: displacements, strains and vibrations [19]. The transverse contraction coefficient, also known as Poisson ratio, is a key elastic constant used for characterizing the mechanical properties of composite materials that can be determined through digital image correlation, which is further needed in the fulfillment of finite element numerical simulations [20]. Nowadays, finite element analyses (FEA) represent the most common method for designing and evaluating the performances of a composite structure [21]. Commercial FEA products have been developed in order to allow a broader range of simulations, which faithfully characterize the specific character of composite materials. Some of the most widespread software programs include: Ansys with the Composite PrepPost analysis (ACP), NX Nastran, Abaqus or Solidworks, which can simulate basically all the major composite categories, such as laminates, foam cores and honeycombs [22].

Carbon fiber reinforced plastics represent an intensely studied category of composite materials, with both tensile [23,24] and flexural [25,26] performance evaluation being at the heart of their characterization. The lack of similar information regarding the linkage between the mesoscopic effects of single-ply CFRP compared to the macroscale mechanical characteristics evaluation was the main reason for addressing this subject.

The main testing procedures covered in the current article are tensile and three-point bending testing, all realized with the aid of the Q400 D.I.C. system from Dantec Dynamics and the INSTRON 8801 servo hydraulic testing system. The objective of the experimental study is to determine the mechanical characteristics of a single-ply CFRP and to assess the relationship with the multi-ply CFRP made of the same material, which will constitute the basis for manufacturing a helicopter tail rotor blade. This shall be realized by examining the correlation between the visible small-scale effects, the displacement and strain position in connection with weave characteristics, and then transposed on the multi-ply composite, which will constitute the skin of the blade. The aerodynamic pressure loading on the tail rotor blade has been validated in previous studies [27]; the tensile pull being caused by the centrifugal forces acting exterior to the rotation axis. Other materials and structural configurations of interest towards developing a composite helicopter blade have also been studied in previous articles [28].

## 2. Materials and Methods

### 2.1. Materials Presentation

The current study represents a comparative approach regarding the mechanical characteristics of a single-ply and a multi-ply CFRP. A total of ten single-ply tensile test specimens were manufactured, each having a rectangular shape and dimensions of approximately 60 mm in length and 10 mm in width. All the single-ply tensile test specimens were dry cut with a diamond disk at the specified dimensions, from a 210 mm × 297 mm CFRP panel. Out of the ten tensile test specimens, five were cut along the warp of the weave (referred to as “U” specimens in the following sections) and the other five were cut along the weft (referred to as “B” specimens).

The tensile and flexural multi-ply specimens, on the other hand, are manufactured by hand lay-up from a number of twelve over stacked plies. It is important to state that the same 0°/90° orientation was employed for all the plies comprising the multi-ply specimens. The manufacturing process consisted of bonding together pairs of successive plies, each measuring 297 mm × 420 mm, by using uniformly distributed epoxy resin and then applying pressure with a hand roller to remove entrapped air bubbles and excess polymer. The multi-ply test specimens were left to cure at room temperature for approximately one week, before being cut to the dimensions specified in Figure 1, with the aid of a waterjet cutting machine. A total of five tensile specimens and three flexural specimens were manufactured for experimental use.

The raw materials used for the experimental study are a carbon fiber fabric with a twill 2/2 weave, embedded in a high resistance epoxy resin. The carbon fabric has the commercial reference GG285T and is manufactured by Castro Composites. The fabric has a bidirectional orientation, with the warp and the weft being disposed at 0° and 90° to each other, resulting in a material density close to 285 g/m^2^. Due to the balanced construction of the fabric, the thread counts per centimeter are about 7.0 and the 3K tow sizes are identical in both directions. With a tensile elastic modulus between 200–280 GPa and a tensile strength higher than 2.5 Gpa, the fibers comprising the fabric can be classified as having a standard elastic modulus type (HT) [29].

The matrix component of the carbon fiber reinforced plastic is an epoxy vinyl ester resin, commercialized under the name Derakane Momentum 470–300, produced by Ashland. As the manufacturer describes in the technical sheet, the resin has a novolac-based vinyl ester design and offers exceptional mechanical properties. The resulting material maintains its toughness and strength at high temperatures, being suitable for CFRP manufacturing. Thanks to its high viscosity, the resin facilitates various manufacturing methods, such as filament winding and also contact molding applications, but can be also used for hand lay-up, spray-up, pultrusion or resin transfer molding.

This combination of carbon fiber fabric and epoxy resin is a preferred choice of composite manufacturers as a consequence of the compatibility they display [30,31], particularly in terms of strength and reduced mass, which are key factors for aerospace components. The main properties of the materials employed in the specimen fabrication are highlighted in Table 1.

### 2.2. Tensile Test Experimental Set-Up and Methodology

#### 2.2.1. Single Ply Tensile Testing

The mesoscopic tensile testing implied the use of the digital image correlation system in its Micro D.I.C. configuration, equipped with two Q400 5-megapixel cameras, mounted on the Leica M125 digital stereo microscope. This arrangement facilitates the experimental visualization of a region of interest no greater than 17 mm, which fits well with the central region of the test specimens. The ten tensile testing probes were each placed under the SK microscope lens, which has focal length of 50 mm and a C.C.D. (charged-coupled device) sensor size of 4/3″ [34]. The uniaxial testing equipment is powered by a stepper electric motor, which can be commanded to move or hold the position of the load cell in any intermediary step. The movement of the electric motor is transmitted through a set of 3D printed gear wheels to the two lateral leadscrews, which translate the rotational motion in a linear motion of the load cell. The device command and control has been realized with an Arduino Uno microcontroller board, equipped with three commanding buttons, which transmit the compression, hold or expansion movement command.

The digital image correlation method is an optical technique used to track surface patterns modifications, eliminating the need for gauges or extensometers, in order to determine the material displacements and the surface strains of a body with six degrees of freedom. By comparison with strain gauges, this optical system offers accurate and reliable information if the set-up and calibration are properly realized. It is estimated that the digital image correlation system has an accuracy of 10% for strains under 0.3%, rising at higher values for strains above 1% or 3% [35].

The two top mounted cameras dispose of a USB 3.0 connection, 75 Hz frequency and a C-mount sensor of 2/3″ diameter. They are an integral part of the Q400 digital image correlation system manufactured by Dantec Dynamics, which has an area measuring capability starting from 5 × 5 mm^2^ up to 1000 × 750 mm^2^. The optical measuring device, in this configuration, has the capability of measuring up to 100% material strain with no direct contact. The main components of the experimental system can be observed in Figure 2.

Before being placed in the grips of the device, white spray particles were dispersed on the surface of the specimens so that a speckle pattern is obtained, in contrast with the typical carbon fiber black color. This represents an essential test condition due to the fact that the measurement process relies on optically tracking the movement of the contrasting points on the specimen. The cross-section area of each tensile test specimen was measured before commencing the tests and was introduced in the ISTRA 4D software used for visualizing the experimental outputs. The software was set up to acquire between 1 and 500 frames, with a frequency of 5 Hz, even though the system has a capacity of up to 560 Hz. The cameras are capable of taking captions with a speed as fast as 47 μs. The tensile speed of the single-ply tensile testing device presented in Figure 3 was set to a value of 2 mm/min.

In order to assure that the material is properly stressed and that the fixing holes are not ovalized, aluminum reinforcements were applied at both sides of the probes, leaving only the center section of the material to be visualized by the two cameras. The specimens in their final form were placed in tensile testing device and were secured with screws on both ends. The microscope disposes of a HILIS cold illumination light, which facilitates the homogenous illumination of the specimens.

#### 2.2.2. Multi Ply Tensile Testing

In order to evaluate the mechanical characteristics of the multi-ply material, which will constitute the aerodynamic skin of a helicopter tail rotor blade, the tensile tests were carried out for a number of five dog-bone shaped specimens. The tension testing was realized using the INSTRON 8801 servo hydraulic system presented in Figure 4, equipped with a digital extensometer, which disposes of a traction force of 100 kN. The device can be employed in both static and dynamic axial testing, for tensile testing, compressive testing, three-point bending and also fatigue testing. The maximum distance between the grips is about 820 mm and the distance between the two vertical columns, which support the load cell is around 455 mm. Conformance with the ASTM 3039 standard for tensile testing of composite materials [36] was followed throughout the entire procedure.

The tensile specimens were placed inside the hydraulic grips of the testing equipment, a preset loading speed of 2 mm/min being utilized. The digital extensometer was placed on the side of each specimen in order to determine the tensile modulus and the elongation of the material.

Results examination and presentation was achieved with the WaveMatrix dynamic testing software. The data acquisition frequency of the software was set to return the measurements once every second for the entire test duration.

### 2.3. Bending Test Experimental Set-Up and Methodology

The three-point bending test method was used for determining the flexural properties of the multi-ply CFRP, by following the guidelines of the EN ISO 14,125 standard [37] used specifically for determining the flexural properties of plastics. This method is especially indicated for composite materials, which do not fail or do not reach yield stress within the 5% strain limit. The main advantage of this testing method is the simplicity in which the specimens are manufactured and tested, and also the localized stress isolation.

The samples utilized for the bending tests have a rectangular profile, being manufactured from a number of twelve carbon fabric plies embedded with the epoxy resin, similar to the bone-shaped tensile specimens. The flexural specimens were placed in the INSTRON 8801 axial testing device equipped with the bend test fixtures, with the extremities positioned on the two vertical support pillars and the force being centrally applied, in the manner presented in Figure 5. The results have been determined by using the digital image correlation system in order to assess the stress–strain evolution of the material.

All the experimental investigations were performed in the Strength of materials Department laboratories of the “Politehnica” University of Bucharest, under following ambient conditions:Room temperature—23°;Relative air humidity—under 60%;Low UV radiation exposure.

## 3. Results

### 3.1. Tensile Performance Analysis of the Single-Ply CFRP Specimens

The single-ply material was subjected to tensile loading with the purpose of assessing its orthotropic behavior with respect to the material axes and to determine its main mechanical characteristics: tensile strength, Young’s modulus and Poisson ratio. Material failure was not the purpose of the tension tests and thus was not achieved during the trials, due to the limited tensile force capability of the testing device. The maximum force applied on each specimen is positioned in the range of 493 N up to 602 N. The mean values obtained for the tensile specimens cut along the warp and weft direction are presented in Table 2.

Based on the previously displayed results, the mean value of Young’s modulus for the warp cut specimens is 24,352.338 MPa on the proportionality portion of the stress–strain curve (σ-ε), with a coefficient of variation of 17%, whilst Poisson ratio has a mean value of 0.351 with a variation of approximately 12%. On account of the inconsistent variation of the U_2_ tensile specimen, this value was not taken into account in determining the mean values previously presented. The maximum real deformation for the warp cut test specimens has a value of approximately 0.46%.

The mean value of the elastic modulus for the weft cut specimens is 23585.489 MPa on the linear elastic region, with a coefficient of variation of 22.25%, while the transverse contraction ratio has a mean value of 0.3495. The B_1_ tensile sample was not considered in determining the mean values, due to unusual mechanical behavior displayed during the experiments. The maximum real deformation for the weft cut test specimens has a value of approximately 0.44%.

The force–displacements evolution for the single-ply test specimens can be classified as predominantly nonlinear, with certain segments of linear elasticity, where the mechanical characteristics could be determined. This is mostly due to the reduced capability of the traction device and the high material strength, specific for CFRP, which makes it difficult to assess the complete stress–strain curve evolution. The linear elastic variations of stress–strain curves for the single-ply tensile test specimens are presented in Figure 6.

Figure 6a exhibits a similar evolution for all of the five warp cut specimens, with the exception of the U_2_ sample, which has a smaller linear section compared to the other four. For this reason, it has been excluded from the mean values determination but was presented graphically for comparison purposes.

The stress–strain evolution posted in Figure 6b is relatively similar in terms of size and shape for all of the five specimens, but with a slight scatter in terms of strain values. The B_2_ specimen is characterized by a steady linear elastic evolution closely after the tensile force was applied, while for the B_1_ and B_5_ test, specimens have a nonlinear behavior during the first few frames, until proper material tension is obtained. These differences are most probably caused by the aluminum reinforcement bonds placed at both ends of the specimens, and the necessary time period for the composite ply to fully take over the applied force. Nonetheless, all of the previously displayed specimens have a similar evolution. It can be stated that the slope of the stress–strain curve for the specimens cut on the warp direction is slightly greater than that of the weft cut specimens, being thus a little stiffer when loaded in this direction, according to the experimental results.

In order to assess the microscopic mechanical behavior of the specimens, certain captions have been presented in the figures below from the image processing software of the digital image correlation system. Figure 7 and Figure 8 exhibit frames from the tensile testing process when the fabric components are placed under visible tension, corresponding to 180 s from the tensile test commencement.

The displacements presented in Figure 7a are with rigid body movement removed (RMBR), in order to better perceive the motion of the carbon fiber filaments. Figure 7b exhibits the manner in which the weft of the carbon fabric is stressed during tensile testing. The fabric weft, which runs across the length of the specimen, is placed under tensile tension (displayed in red), while the warp filaments remain in a more relaxed posture. The obtained visual effect for all of the tested specimens is that of two alternative diagonal patterns. This effect is also present in the specimens presented in Figure 8 the warp filaments being, this time, placed on the tensile direction. A mirror effect can be visualized in comparison with the weft cut specimens, with the presence of a similar diagonally shaped pattern specific for the twill 2/2 weave. The noncolored points from the previous images signify the nodes, which could not be traced by the software when analyzing the caption frames in order to detect the filament motion. In other words, the displacement in the Z direction of the specimen illustrates the tendency of the weave to pull the material towards the horizontal symmetry axis, generating a stiffening effect.

The strain characteristics of the specimens reveal a similar mechanical behavior. It is observable in Figure 8 that the maximum strain values occur on the fibers being placed on the tensile direction, revealing a diagonal strain pattern, similarly with the manner in which the fibers are intertwined on the right side of the figure. Small matrix cracks occurred during the testing procedure for the specimens, which surpassed the 85 MPa tensile strength of the epoxy resin, but they were not visually identifiable under the microscope lens. Overall, the high resistance of matrix made the tensile test very accessible for the material. At an approximate 100 MPa maximum measured tensile stress, the carbon fiber was 44 times below the maximum stress capacity specified by the manufacturer.

### 3.2. Tensile Performance Analysis of the Multi-Ply CFRP Specimens

The experimental tests were realized in accordance with the ASTM D3039 standard, the tensile specimens being stretched to the point of failure. The stress–strain dependence of the five multi-ply specimens is graphically presented in Figure 9.

The previous diagram clearly describes a bilinear mechanical behavior, in which the material starts with a linear elastic portion, which is preceded by a plastic segment of a smaller length. The mean longitudinal elastic modulus corresponding to the 0.05–0.15% specimen strain is represented by the slope of the linear elastic portion at a value of 42 GPa. This value, also known as the constant of proportionality for the region in which the stress is directly proportional to the strain, is valid up until the yield strength of the material is reached, at a value of about 448 MPa. With a standard deviation of only 4.23%, the specimens exhibit a similar mechanical behavior along the length of the tryouts. The upper elastic limit marks the passing to a linear plastic behavior. The modulus of elasticity determined for this plastic region, also known as tangent modulus, was determined for the 1.1–1.3% strain segment with a mean value of 19.457 GPa. The failure stress has a mean value of 480.57 MPa, being determined for each individual sample by dividing the applied tensile force to the cross-section area of each specimen. The failure mechanism is marked by a sudden material rupture, with no identifiable necking phenomenon. All the determined mechanical characteristics are tabularly presented in Table 3.

The transverse contraction coefficient has a value of 0.353 and has been determined solely for the fifth tensile test specimen analyzed with the digital image correlation technique. This is due to the fact that the digital extensometer used with the INSTRON tensile testing equipment measures only the uniaxial strain and has no record of the transverse strain. The fifth specimen tensile modulus has a value of 41.525 GPa for the 0.05–0.15% strain segment, in close proximity with the median value determined for the first four tensile specimens. The tensile specimen evaluation using the digital image correlation system is presented below in Figure 10.

The tendency of the stress to perpetuate with respect to the direction of the tensioned fibers is visible in a similar matter as with the single-ply tensile tests, the diagonal pattern being clearly visible.

All material failure occurred in the reduced section of the tensile test specimens, making them valid in terms of conformance with the tensile testing standard. The failure mechanism implied, in the first stage, the weakening of the adhesive bond between the material plies and also between the resin and the carbon fabric. The final material failure is characterized by a generalized delamination on the central section of the specimens, with local fabric ruptures, a common failure mode of carbon fiber-reinforced composites [38]. The carbon fiber rupture occurred only as a small local effect due to the fact that the tensile stress at the breaking point was nearly ten times lower than the tensile strength declared by the manufacturer. A detailed view of the failure location can be seen in Figure 11 for three of the five multi-ply tensile test specimens, where the failure was more visible.

### 3.3. Flexural Performance Analysis of the Multi-Ply CFRP Specimens

The bending tests were realized for three specimens with a rectangular section, by using the three-point bending test procedure. The results have been evaluated using the DIC method, which implied painting the specimens with black dots on a contrasting white background on the area facing the cameras, to easier identify the material movement. The bending test is generally characterized by a bending moment, which acts on the transverse section of the specimen, with a vector perpendicular to the specimen central axis. In general, the bending moment is also accompanied by a shear force. It is important to state that the three-point bending mechanism stretches the fibers located on the lower side, making the material thinner and compresses the upper side of the material, hence increasing its thickness.

Hooke’s law for linear elastic materials subjected to bending forces has the following form, for a specimen with its length in the X axis direction:(1)$$\sigma_{x}=E\cdot \varepsilon_{x}=\frac{E}{\rho_{z}}\cdot z=k_{y}\cdot E\cdot z$$
(2)$$\sigma_{x}=E\cdot \varepsilon_{x}=\frac{E}{\rho_{z}}\cdot z=k_{y}\cdot E\cdot z$$
where, *E* is the elasticity modulus, *k_y_* is the curvature of the deformed mean fiber and *z* is the distance from the mean fiber.

Representative sections from the bending tests have been presented in Figure 12, where captions from the interpreting software were taken to highlight the displacements on the Y axis, perpendicular to the specimen length, with rigid body movement removed.

The ply delamination can be spotted in the previous figure due to the absence of the color scale, determined by the sudden local material deformation. The ply rupture can be observed near the application of the shear stress. The progressive movement of the central beam produces tensile stress in the lower half of the specimen and compressive stress in the upper half, both being delimited by a neutral central plane. The graphical evolution of the Y axis displacements with the force applied is depicted in Figure 13 for the three tested specimens.

The three specimens have a similar general evolution, with a linear slope up to the maximum force applied, consisting of a mean value of 0.5966 kN. For the first two tests, the force–displacement evolution maintains a constant level after the maximum force was reached, after which the force declines and material failure occurs.

The flexural modulus of elasticity, designated with the notation E_flex_, is determined for the linear elastic segment of the force–displacement curve, according with the mathematical expressions provided by the standard EN ISO 14,125 [31], as follows:(3)$$E_{flex}=\left(L^{3}/4\cdot b\cdot h^{3}\right)\cdot \left(\Delta F/\Delta v\right)$$
where: *L* is the distance between the base simple support, equal to 70 mm; b and h are the dimensions of the rectangular cross section.

Similarly, the flexural stress and strains are determined with the aid of the following mathematical relations:(4)$$\sigma_{f}=\frac{3\cdot F\cdot L}{2\cdot b\cdot h}$$
(5)$$\varepsilon =\frac{6\cdot s\cdot h}{L^{2}}$$

The main flexural characteristics of the material are presented in Table 4, calculated for each individual specimen with the equations previously presented, in accordance with the EN ISO 14,125 standard for the determination of flexural properties of fiber-reinforced plastic composites.

The state of the specimens after failure can be observed in Figure 14 listed below. The failure propagation is visible with the naked eye, at the location where a linear discontinuity appears on the visible painted surface, near the application point of the central force.

The main identifiable failure mode is delamination of the bonded plies near the central load application position, also accompanied by ply rupture in the case of one probe.

### 3.4. Finite Element Analysis of the Multi-Ply CFRP Specimens: Results Comparison

The finite element analyses were realized with the purpose of replicating the experimental conditions and results, by defining the composite material in the FEA software with the main mechanical characteristics determined in the previous subchapters. By studying the multi-ply tensile test results, we can clearly state that the material has a multilinear isotropic hardening behavior. Therefore, in order to realize an FEA simulation of the specimens, the characteristics for both the elastic and the plastic isotropic region have to be precisely defined. The linear elastic region is characterized by the slope of the stress–strain curve, meaning Young’s modulus and also Poisson’s ratio, which has been determined using the digital image correlation method. The necessary characteristics for the plastic deformation region, which must be inserted in the Ansys Workbench simulation software were determined from the engineering results, by calculating the true stress and strain values, using the equations stated below [39]:(6)$$\sigma_{\mathrm{true}}=\sigma_{\mathrm{eng}}\cdot \left(1+\varepsilon_{\mathrm{eng}}\right)$$
(7)$$\varepsilon_{\mathrm{true}}=\ln\left(1+\varepsilon_{\mathrm{eng}}\right)$$

The plastic strain of the material is determined by subtracting the elastic strain from the total strain as depicted below:(8)$$\varepsilon_{plastic}=\varepsilon_{total}-\varepsilon_{elastic}$$
where the elastic strain is determined using the following formula:(9)$$\varepsilon_{elastic}=\frac{\sigma_{\mathrm{true}}}{E}$$

The simulations were performed using an explicit dynamics analysis, in order to obtain a relevant insight of the material performance. The material layup of twelve overlapping plies and the overall orientation was introduced using the Ansys ACP composite set-up. Both tensile and bending specimens resulted in an overall thickness of 3.96 cm, the equivalent of 0.33 cm per individual ply. The fiber orientation is defined by placing the warp perpendicular on the weft of each ply, the orientation remaining constant through the entire material.

The specimens were modelled as shell surfaces with quadrilateral elements, which resulted in approximately 2104 surface elements for the tensile specimen and 800 elements for the bending specimens. The surface mesh was realized using an element size of 1 mm, in order to obtain a sufficient amount of detail in the stress and strain diffusion. The element distribution on the finite element modeled specimens can be observed in Figure 15a,b.

The tensile specimen model was fixed on one end in order to eliminate all motion, while on the other end a displacement was applied as Figure 16b depicts. The flexural specimen on the other was simply supported on one end and provided with a null displacement on the Z axis direction at the other end, as to reproduce the practically occurring state. The displacement was applied on the central middle section of the specimen, as the mean value that occurred in the experimental tests (Figure 16a).

The finite element analysis was realized in order to assess the resulting stress and strain, in comparison with the experimental tests. With regard to the flexural analysis, the directional deformation of the vertical direction can be seen in Figure 17a, and the normal stress distribution on the specimen direction can be visualized in Figure 17b.

Figure 17 also presents the finite element analysis results obtained for tensile test specimen loaded on the longitudinal direction, in the explicit dynamics analysis: the total deformation of the specimen (Figure 17c) and equivalent von-Mises stress (Figure 17d).

In Table 5, a comparison between the experimental test results and finite element analysis results is presented, with respect to Young’s modulus of elasticity for the tensile test and the flexural modulus for the three-point bending test. The accuracy of the results is evaluated by determining and assessing the differences in values. The elasticity moduli for both FEA analyses was obtained as the slope of the stress–strain curve determined by the numerical software.

## 4. Conclusions

The paper reports the tensile and flexural mechanical properties obtained for the carbon fiber-reinforced material, in both single-ply and multi-ply test configurations. The results offer an additional contribution to the study of carbon fiber-reinforced plastics with epoxy resin matrix, creating a link between the mesoscopic and the macroscopic mechanical behavior of the material.

The single-ply experimental results lead to the conclusion that the two specimen types have similar characteristics in both weft- and warp-loading direction, with a standard deviation of only 0.5 GPa and a mean value of 23.968 GPa for the tensile modulus. The examination of the small-scale fiber loading realized with the Micro-DIC setup correlates well with the large-scale mechanical behavior, offering a better understanding of the material failure. Stress and strain properties measured with the DIC Q400 system are in range with the values obtained from the INSTRON 8801 load cell, thus being reliable and expanding the overall data set.

The ratio of the transverse strain to the axial strain is accurately determined by using the DIC method and has similar values for both single-ply and multi-ply specimens, below the value of 0.5 for isotropic material.

The elastic properties obtained for the CFRP are used to properly define the material characteristics for the numerical analysis. Tensile and flexural finite element simulations have been realized, in order to assess the equivalence between the experimental and numerical results. The reduced error of the elastic modulus for both tensile and flexural simulations implies that the FEA model is trustworthy and can be further used in developing composite structures for aeronautic purposes.

## Figures and Tables

**Figure 1 polymers-14-03213-f001:**
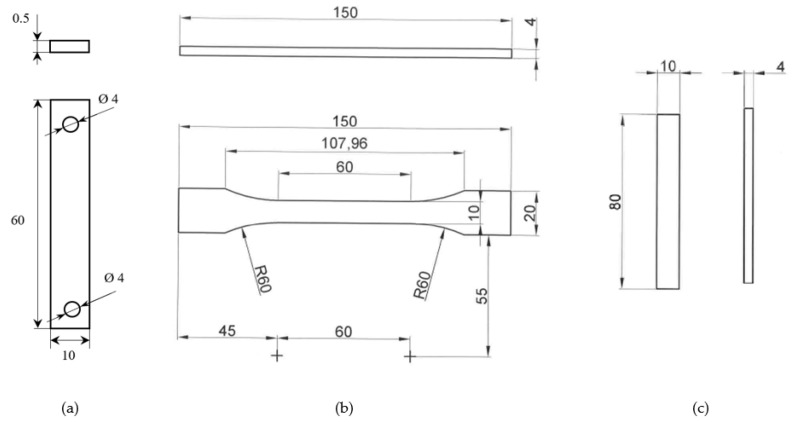
Dimensions in millimeters of the tensile test specimens: (**a**) single-ply tensile specimen, (**b**) multi-ply tensile specimen, (**c**) multi-ply bending specimen.

**Figure 2 polymers-14-03213-f002:**
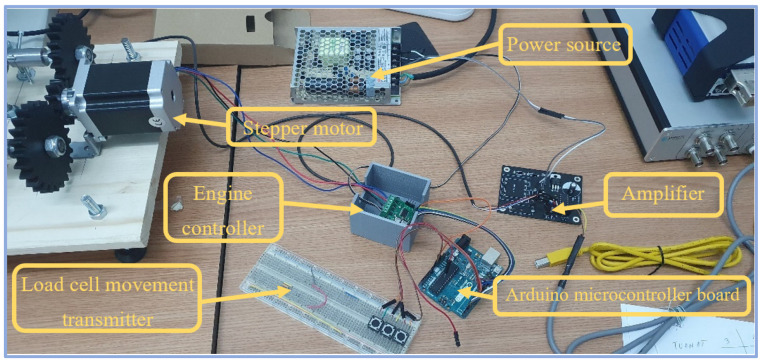
Main components of the experimental set-up for the single-ply tensile testing.

**Figure 3 polymers-14-03213-f003:**
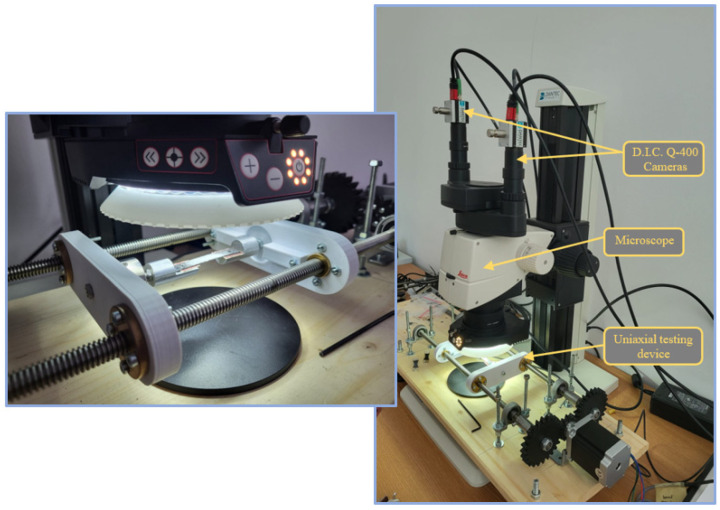
Visualization equipment placed over the specimen fastened in the uniaxial testing device.

**Figure 4 polymers-14-03213-f004:**
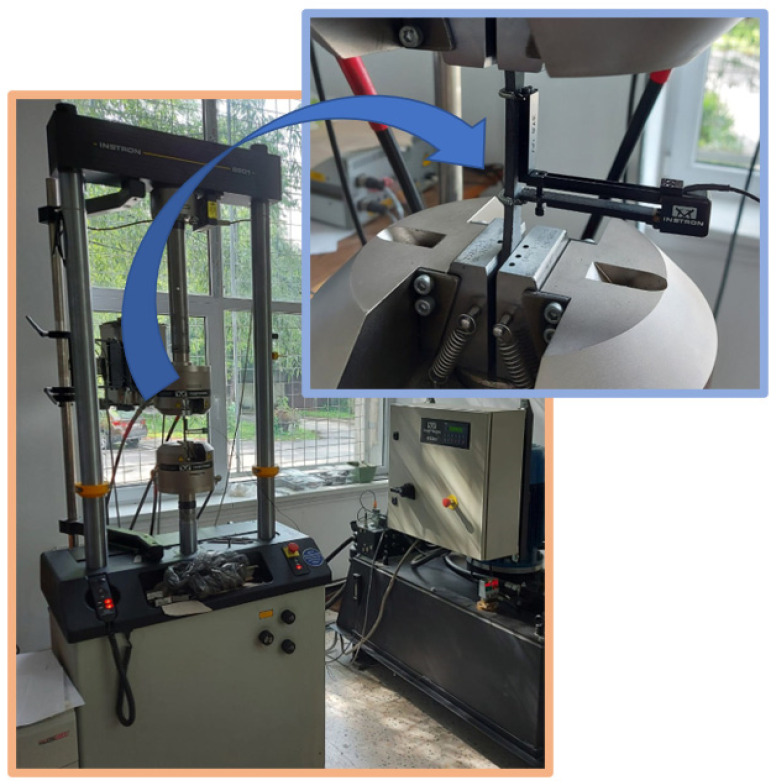
Tensile testing of multi-ply CFRP.

**Figure 5 polymers-14-03213-f005:**
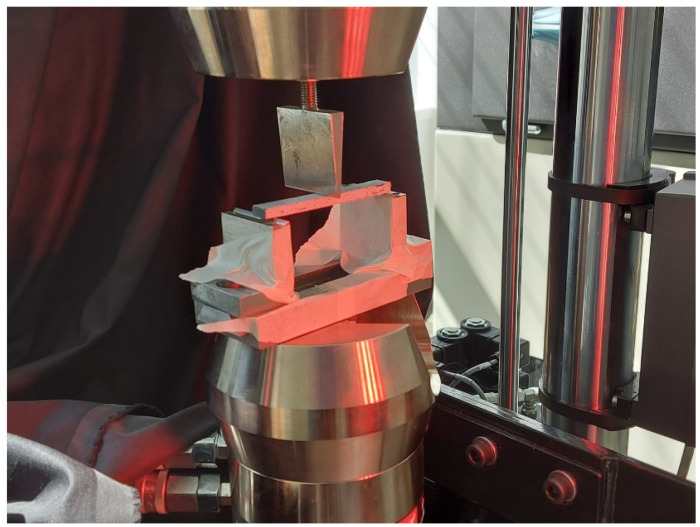
Bending test of multi-ply CFRP.

**Figure 6 polymers-14-03213-f006:**
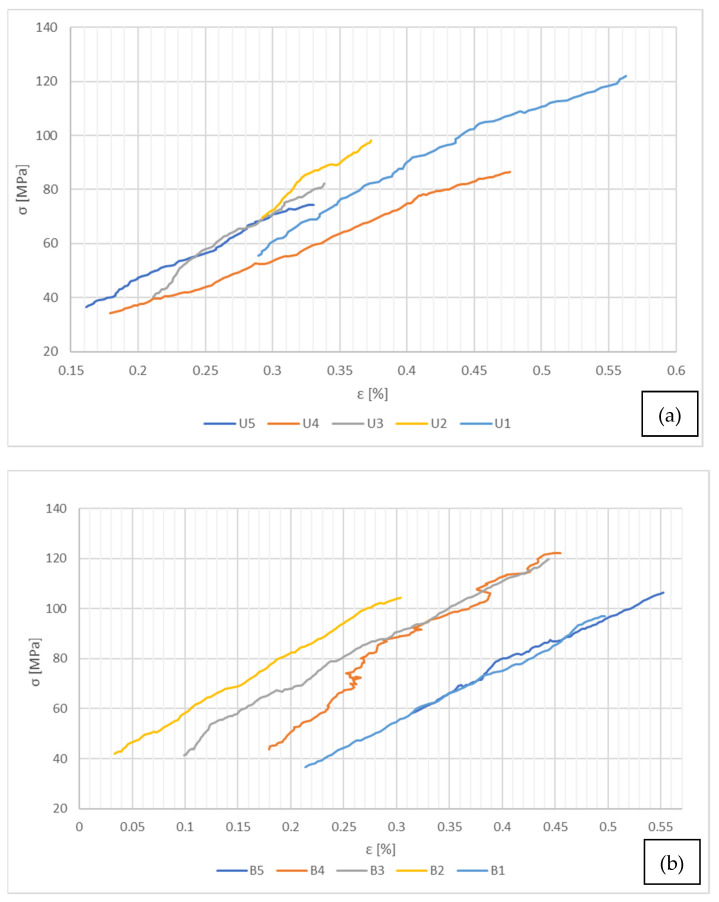
Linear elastic variation for the single-ply test specimens—(**a**) warp cut specimens, (**b**) weft cut test specimens.

**Figure 7 polymers-14-03213-f007:**
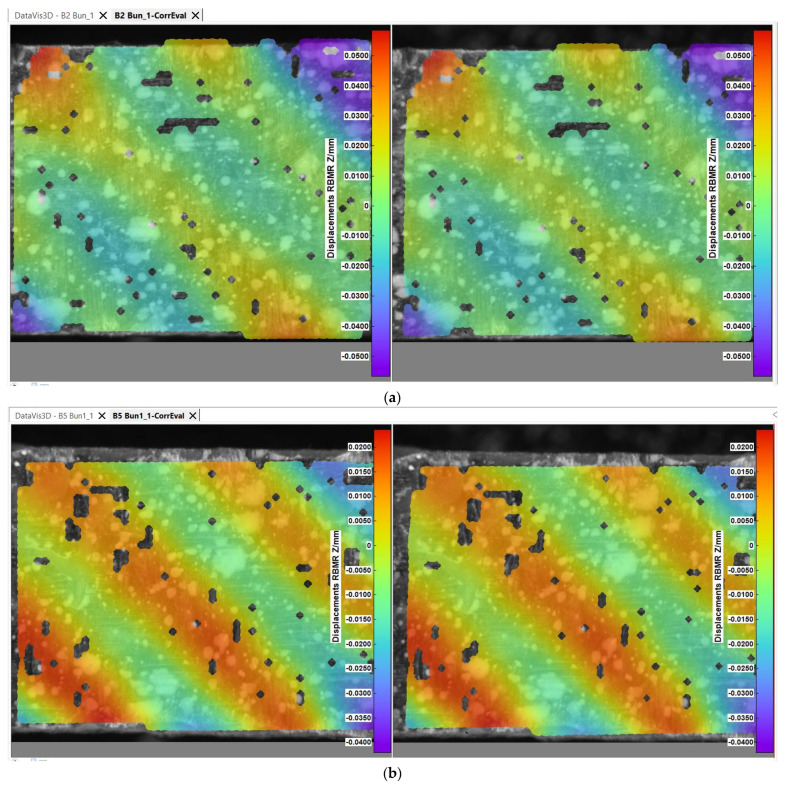
Displacements on Z direction for the single-ply test specimens—(**a**) B_2_ specimen, (**b**) B_5_ specimen, (**c**) U_4_ specimen, (**d**) U_5_ specimens—test time 180 s.

**Figure 8 polymers-14-03213-f008:**
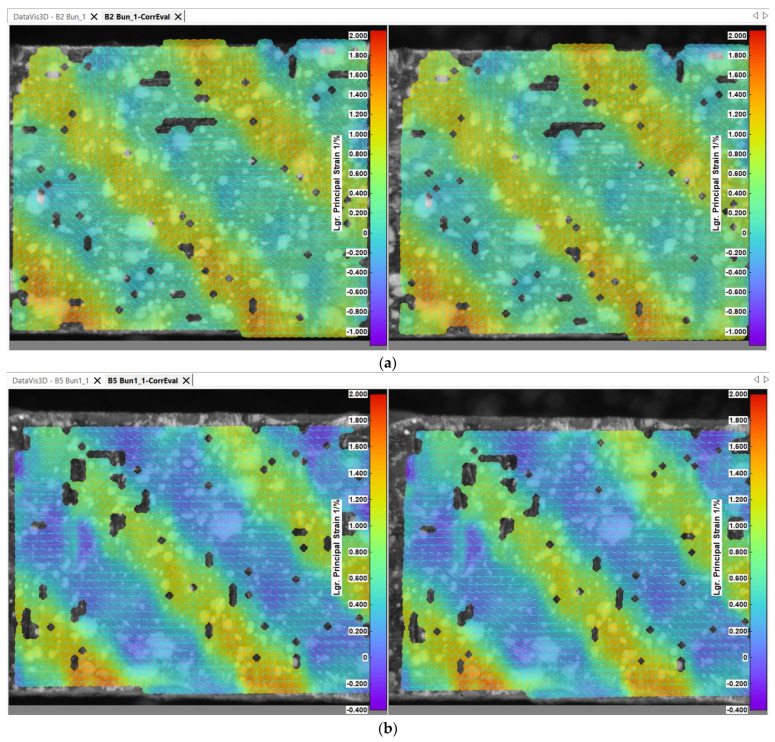
Principal strain for the single-ply test specimens—(**a**) B_2_ specimen, (**b**) B_5_ specimen, (**c**) U_4_ specimen, (**d**) U_5_ specimens—test time 180 s.

**Figure 9 polymers-14-03213-f009:**
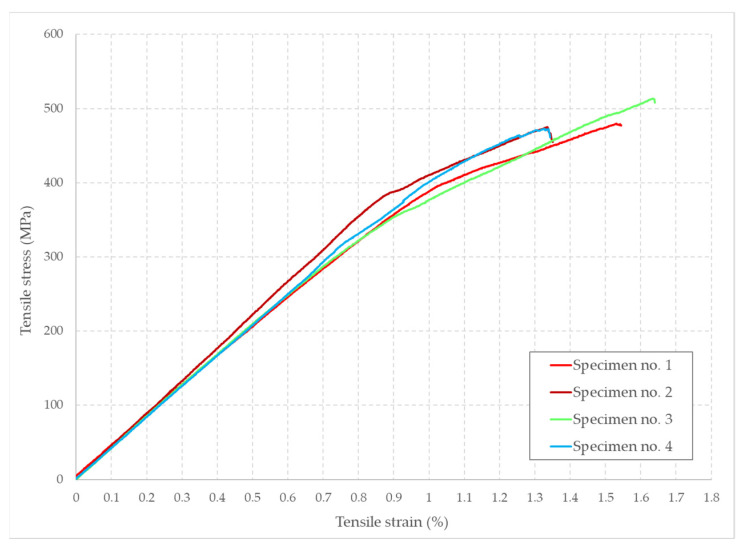
The stress–strain diagram for the multi-ply test specimens.

**Figure 10 polymers-14-03213-f010:**
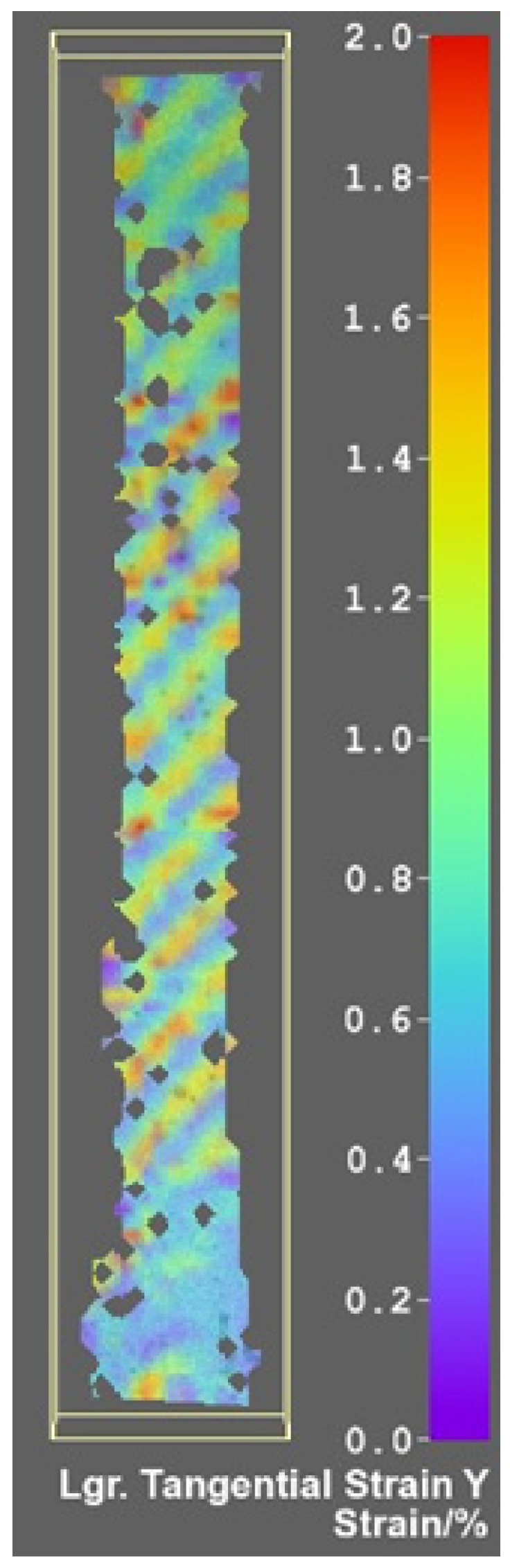
Maximum strain obtained for D.I.C. evaluated multi-ply tensile test specimen.

**Figure 11 polymers-14-03213-f011:**
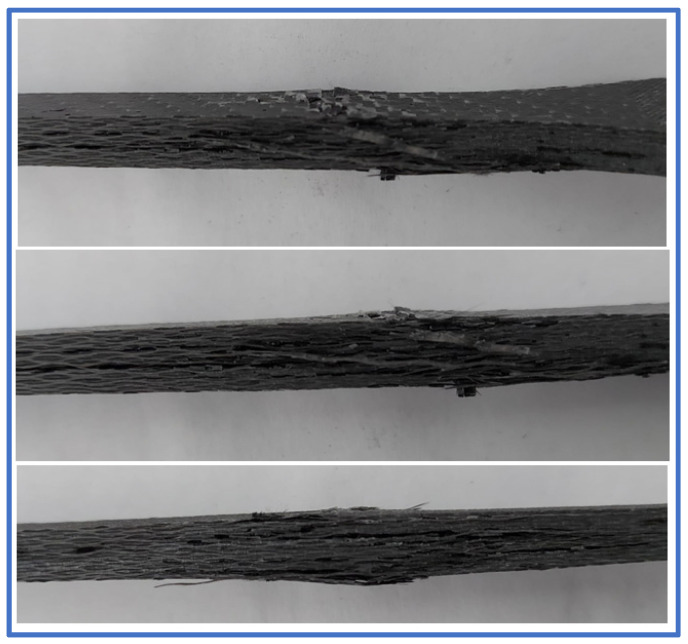
Material failure localization on the tensile test specimens.

**Figure 12 polymers-14-03213-f012:**
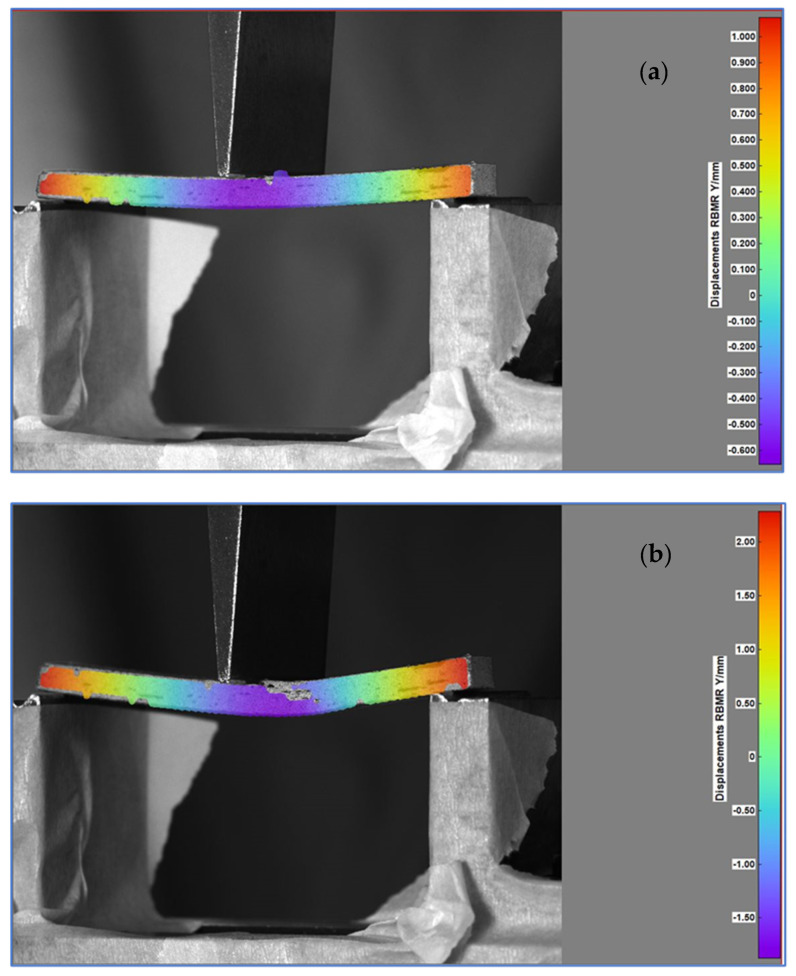
Specimen subjected to bending load, midway—Frame 600 (**a**) and near the end—Frame 1200 (**b**) of the test.

**Figure 13 polymers-14-03213-f013:**
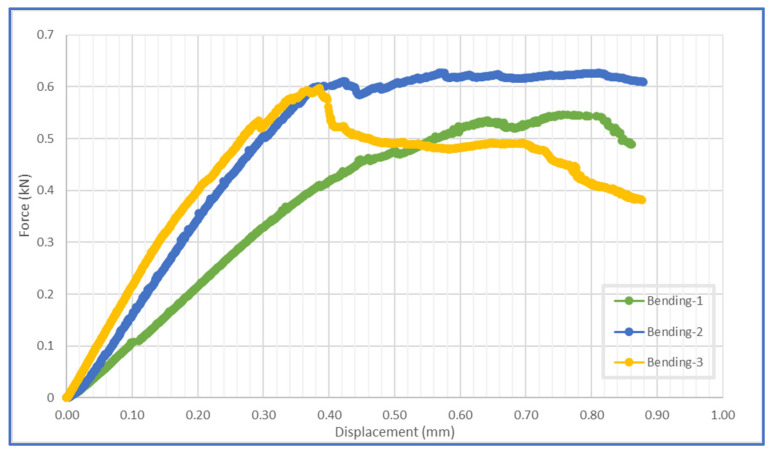
Force–displacement evolution for the bending tests.

**Figure 14 polymers-14-03213-f014:**
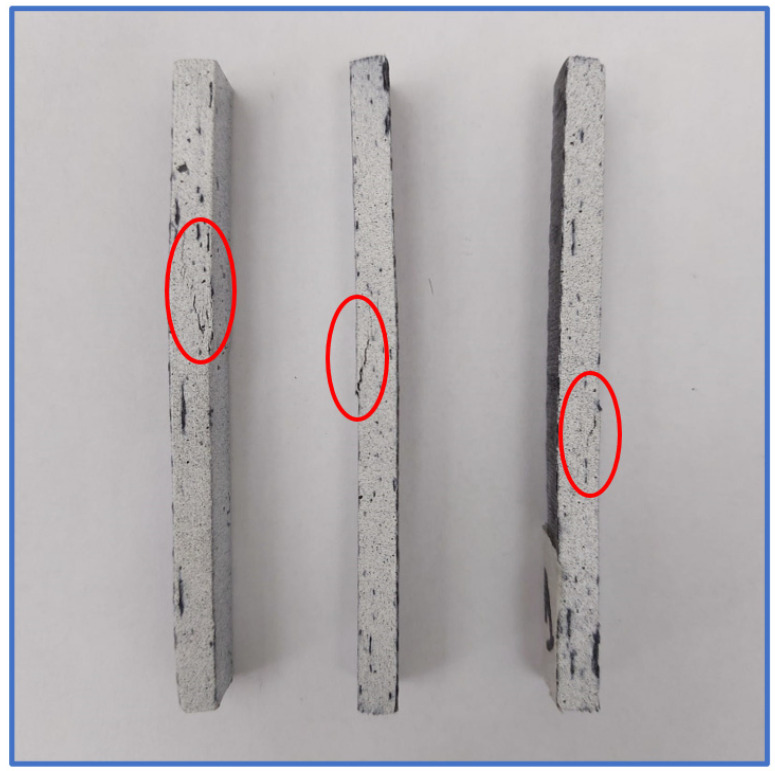
Bending test specimens after testing.

**Figure 15 polymers-14-03213-f015:**
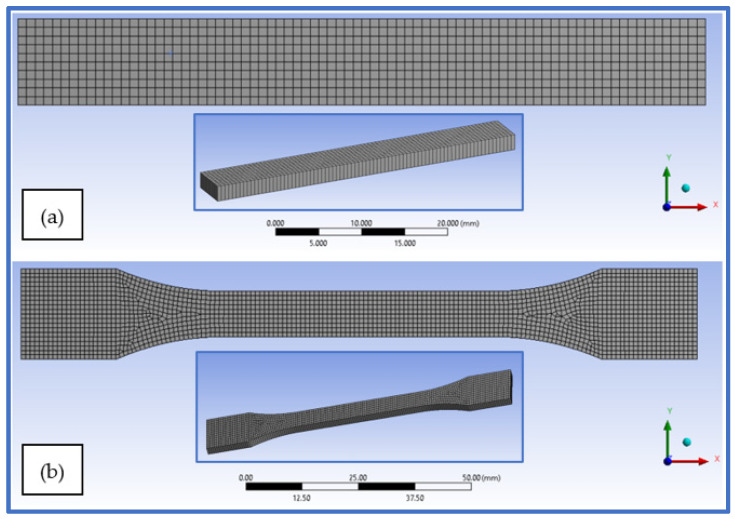
Hex-dominant mesh of the finite element specimen models—(**a**) bending specimen, (**b**) tensile specimen.

**Figure 16 polymers-14-03213-f016:**
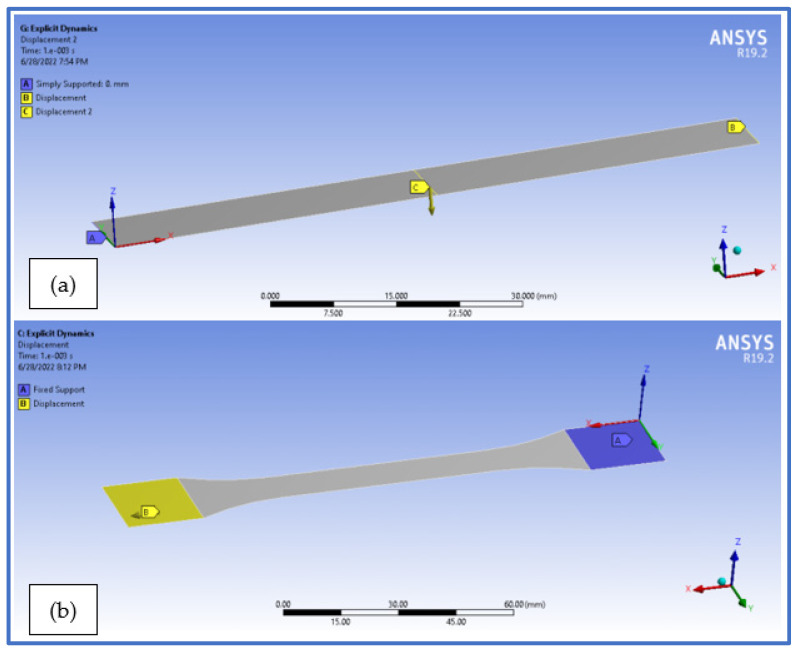
Analysis setup for the finite element specimen models—(**a**) bending specimen, (**b**) tensile specimen.

**Figure 17 polymers-14-03213-f017:**
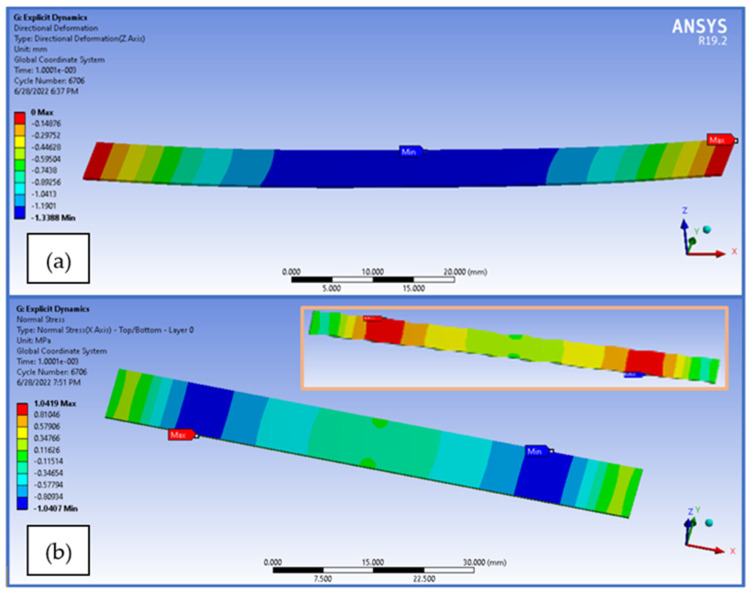
FEA results obtained for the tensile and flexural tests—(**a**) flexural test vertical displacement, (**b**) normal stress on specimen during flexural test (**c**) tensile test total deformation and (**d**) tensile test equivalent (von-Mises) stress.

**Table 1 polymers-14-03213-t001:** Mechanical characteristics of CFRP components as stated by the manufacturers [32,33].

Epoxy Vinyl Ester Resin		3K Carbon Fabric	
Commercial name	Derakane Momentum 470–300	Commercial name	Carbon fabric GG285T
Producer/Distributor	Ashland	Producer/Distributor	Castro Composites
Dynamic Viscosity	325 mPa∙s(cps)	Mass per unit area	285 g/m^2^ ± 5%
Kinematic Viscosity	300 cSt		
Styrene Content	33%	Weave	twill 2/2
Density	1.08 g/cm^3^	Laminate thickness	0.28 mm ± 2.5%
Tensile Strength of casting	85 MPa	Fiber description	HR carbon fiber 3K—200 tex (both weave and weft)
Tensile Modulus of casting	3600 MPa	Thread Count	7 ends/cm (both weave and weft)
Casting density	1.17 g/cm^3^	Tow tensile strength	4410 MPa
	entry 2	Tow tensile modulus	235 GPa
		Typical density	1.79 g/cm^3^

**Table 2 polymers-14-03213-t002:** Mechanical characteristics of the single-ply test specimens.

Specimen Number	Dimensions of Cross-Section		Young’s Modulus [GPa]	Poisson Ratio
	b (mm)	h (mm)		
U_1_	8.84	0.46	23.669	0.460
U_2_	9.01	0.55	33.521	0.390
U_3_	10.43	0.53	31.708	0.204
U_4_	9.62	0.52	18.640	0.290
U_5_	12.90	0.56	23.391	0.450
Median value			24.352	0.351
B_1_	8.86	0.51	21.145	0.520
B_2_	9.05	0.51	23.749	0.310
B_3_	9.63	0.49	21.542	0.320
B_4_	9.54	0.46	29.171	0.398
B_5_	9.15	0.51	19.878	0.370
Median value			23.585	0.349

**Table 3 polymers-14-03213-t003:** Mechanical characteristics of multi-ply test specimens evaluated using INSTRON 8801.

Reference	Dimensions of Cross-Section		Tensile Modulus (Segment 0.05–0.15%) [GPa]	Tensile Modulus (Segment 1.1–1.3%) [GPa]	Tensile Stress at Yield (Offset 0.2%) [MPa]	Tensile Stress at Tensile Strength [MPa]	Tensile Strain at Tensile Strength [%]
	b (mm)	h (mm)					
Specimen no. 1	9.64	4.93	39.837	15.426	440.17	474.24	1.50
Specimen no. 2	9.95	4.8	44.097	19.841	457.18	470.64	1.31
Specimen no. 3	9.43	4.95	41.709	22.025	424.99	506.56	1.60
Specimen no. 4	9.47	4.86	41.288	20.537	471.95	470.86	1.31
Median value			41.733	19.457	448.58	480.57	1.43
Standard deviation			4.24	14.58	4.55	3.62	10.17

**Table 4 polymers-14-03213-t004:** Flexural characteristics of three-point bending tested specimens.

Reference	Dimensions of Cross-Section		Flexural Modulus Linear Elastic Region [GPa]	Maximum Force (N)	Maximum Stress (MPa)	Maximum Displacement at Maximum Force (mm)
	b (mm)	h (mm)				
Bending-1	10.41	4.99	61.035	545.98	1.10	1.82
Bending-2	10.38	4.87	101.011	627.74	1.27	1.27
Bending-3	10.45	5.02	114.590	597.88	1.21	0.92
Median value			92.210	590.53	1.19	1.34

**Table 5 polymers-14-03213-t005:** Comparison between the experimental and FEA analysis results for the tensile and bending tests.

Tensile Modulus of Elasticity [GPa]			Flexural Modulus of Elasticity [GPa]		
Experimental (Median Value)	*FEA*	Error (%)	Experimental (Median Value)	*FEA*	Error (%)
41.733	38.658	7.36	92.21	94.56	2.54

## Data Availability

Not applicable.

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
