# Peer review of "Mechanical Characteristics Evaluation of a Single Ply and Multi-Ply Carbon Fiber-Reinforced Plastic Subjected to Tensile and Bending Loads"

_polymers, 2022, doi:10.3390/polym14153213_

Round 1

Reviewer 1 Report

This paper studies the mechanical characteristics evaluation of a single ply and multi-ply CFRP subjected to tensile and bending loads. The authors have conducted a wealth of experiments and simulation studies. Some results are critical for evaluating the mechanical properties and mechanism of CFRP. However, it still needs further improvement as for writing. Please see the following comments. 

1# Abstract, please provide more qualitative and quantitative results for the current research work. In addition, please indicate the agreement between the experimental results and FEA analysis.

2# Composite are a general term of a class of materials, including many material examples. This paper only mainly focuses on fiber reinforced polymer composites. Therefore, it is recommended to summary the types of materials, performance, advantages and main application fields on fiber reinforced polymer composite. Please review the following latest research on performance evaluation of FRPs, such as mechanical properties, fatigue and durability resistances. Composite Structures, 2020; 246: 112418. Materials and Structures, 2020, 53: 73.

3# Digital image correlation (DIC) is a very effective testing technology to monitor the mechanical properties of FRP. Therefore, some relevant examples about the analysis of mechanical properties through DIC should be added. For example, Composite Structures, 2019, 212: 43-57. https://doi.org/10.1080/15376494.2021.1974620.

4# The logical writing of introduction should be further improved. Too many paragraphs are included in the content. It is suggested that the authors combined them according to the content related to this work, including research background, basic information, advantage and application of FRP, relevant summaries of mechanical properties of FRP, DIC testing technology and examples, FEA and the present work.

5# Part 2.1, Materials presentation, it is recommended to write them according to the raw materials and sample preparation. For raw materials, please provide some basic properties, manufacturers and other sources. For sample preparation, please provide the sample preparation process and method.

6# Part 2.2, as for the tensile test and methods, it is suggested to further refine the descriptions of the test methods and process. Some basic information and advantage about the instruments are suggested to be put in the introduction. At present, it is difficult to provide readers with a clear logical idea owing to too many paragraphs.

7# In the part of “Results”, all the subtitles are inappropriate, because it is not a testing, but the tensile property analysis. It is suggested to uniformly modify them to tensile/bending performance analysis of single-ply/multi-ply composites.

8# In Table 2, Young’s modulus should be represented by “GPa”, and attention should be paid to the number of decimal places and significance for different variables. A similar situation applies to table 3.

9# It is suggested to delete the black background in Figures 7 to 8.

10# From Figure 9 to figure 12, please indicate what is the working condition of DIC test results? What is the test time? Is it a picture of the material at the moment of fracture?

11# The definition of Figure 13 and 17 is not enough. It is suggested to replace it with high-definition pictures.

12# The fourth part should be the conclusion. It is suggested that the authors rewrite the conclusion, including 3~4 key points according to the current research results.

Reviewer 2 Report

The article is very interesting. It concerns research on the strength of composites. The introduction provides a good background to the set goal. Analyzing the literature is up to date. Tables and figures are correctly described. The adopted research methodology is interesting and the results allow to formulate interesting conclusions. The authors demonstrated the good technique of the researcher. The conclusions are correctly formulated.

Such a little bit of a note: Figure 1 is the same as ISO 527-2, it is not necessary to repeat the drawing from now. In the case of composites, it would be worth considering a different shape of the samples. It would be worthwhile to read the individual sheets of the ISO 527 standard in detail.

Reviewer 3 Report

Anton et al studied the study on interesting properties on mechanical behavior of single ply and multi-ply CFRP which is performed under different tensile and bending loads. However, the paper is not well written, with lot of confusion even though supported with lot of experimental results.   Therefore, the paper especially the language need to be polished significantly before publication. Few suggestions are as follows –

[1] The English language need to be edited significantly by expert after revisions to make the paper publishable.

[2] From the result and discussion section, it is not clear what do author want to say and the interpretation is highly confusing. Please try to improve this paper.

[3] Please expand “CFRP” in the title of the paper. The abstract can be further improved by adding few lines on the outcome of the experiments such as improvements in “%”.  Please avoid general lines in abstract that are well known.

[4] Introduction can be improved further. Please refer 3-4 paper from Polymers-MDPI on subject of the paper since there are many papers in subject of the work. Please refer them and explain the novelty of present work on existing literature of the Polymers in last paragraph of the introduction of paper.

[5] Please do figure grouping and make the interpretation simple and informative.  Such as Figure 3-6 into 1 figure and provide the components of Figure 4 and 5 Then, Figure 7 and 8 into 1 figure. Why the data is not starting from 0 in Figure 7-8? Then. Figure 9-10 into 1 figure and Figure 11-12 into another figure. What is the difference between Figure 9-10 & Figure 11-12 need to be addressed clearly? Figure 15-16 should be 1 figure. Figure 19-20 and Figure 21-22 should be merged into 2 figures.

[6] Conclusion is missing? Please provide it? Please provide why this work is important? Experimental outcomes and how this work adds to the literature if published? Good Luck for revisions!

Round 2

Reviewer 1 Report

It can be accepted in the present form. 

Reviewer 3 Report

The language used in the work is still confusing. Please imporve it